

# Annual hydrographic variability in Antarctic coastal waters infused with glacial inflow

Maria Osińska[1], Kornelia A. Wójcik-Długoborska[2], Robert J. Bialik[2]

[1]Institute of Oceanography, University of Gdańsk, Piłsudskiego 46, 81-378 Gdynia, Poland

[2] Institute of Biochemistry and Biophysics, Polish Academy of Sciences, Pawińskiego 5a, 02-106 Warsaw, Poland

*Correspondence to*: Robert J. Bialik (rbialik@ibb.waw.pl)

**Abstract.**

During the thirty-eight months between December 2018 and January 2022, multiparameter hydrographic measurements were taken at thirty-one sites within Admiralty Bay, King George Island, Antarctica. These records consisted of water column measurements (down to 100 m) of temperature, conductivity, turbidity, and pH as well as the dissolved oxygen, dissolved organic matter, chlorophyll A and phycoerythrin contents. The sites were chosen due to their variable distances from glacial fronts and open ocean waters. Fifteen sites were localized within smaller glacial coves, with waters highly impacted by glacial infusions; seven sites were located in the open waters of the main body of Admiralty Bay; and nine were located in intermediate conditions of the Ezcurra Inlet. The final dataset consists of measurements carried out over 142 separate days, with an average 3.74 measurements per month. However, data were not collected regularly throughout the year and were collected less frequently during winter, though data were gathered for all but two winter months. On average, each site was investigated 98.2 times. Due to calibration issues, absolute values of optically measured properties occasionally show impossible negative values, but the relative distributions of these values remain valid. Variabilities in the measured properties each season and throughout the whole duration of the project reveal regular oscillations as well as possible long-term trends.

## 1 Introduction

Fjords and bays mixed with waters from glacial outflow are unique environments that are vital for maintaining polar ecosystems and, considering their size, are critical regions of the global ocean. When freshwater from glaciers is introduced into marine environments, the combination of water masses alter the properties of the seawater, forming glacially modified water (GMW) (Straneo, 2012). This alteration of the ocean's water chemistry has far reaching effects that have been investigated in numerous studies across a diversity of fields. GMW influences the hydrodynamics and thermodynamics of the ocean (Bendtsen et al., 2015; Chauché et al., 2014), changes the ocean's chemical composition (Kanna et al., 2020; Fransson et al., 2015) and impacts ecosystems both directly and indirectly (Gerringa et al., 2012; Oliver et al., 2018). Therefore, there





are significant justifications to investigate water quality properties in glacial bays and fjords and to track their variability to potentially predict future changes.

The majority of studies examining the influence of GMW on seawater have been performed in the Northern Hemisphere. Some notable works in this field have been performed in the Antarctic region (among others: Cape et al., 2019; Forsch et al., 2021; Meredith et al., 2018; Monien et al., 2017; Schloss et al., 2014); nevertheless, widely available data that describe water quality in glacial bays beyond seasonal timescales, at high sampling resolutions, and that examine multiple variables remains non-existent. In fact, such datasets are scarce for the Arctic and Alaska as well.

To address this deficiency, an intricate investigation campaign was designed with the intention of comprehensively observing the seasonal oscillations and long-term trends in water quality variability of Admiralty Bay (AB), King George Island in Western Antarctica. The goal of this project was to widen the scope of previously gathered observations by expanding the overall duration of monitoring, increasing the frequency and number of measured parameters and to acquire data across all seasons of the year.

## 40   2 Research Area

AB is a 177.04 km$^2$ cove southeast of King George Island, the largest island of the Southern Shetlands in Western Antarctica. In its interior, AB is subdivided into three distinct areas: Ezcurra, Mackellar and Martel Inlets, which all blend together approximately 11 km from the open ocean waters of the Bransfield Strait, forming the main body of AB (Figure 1). Its coastline has a length of 150 km, of which 102 km consist of rocky coastline and the remaining 38 km consist of ice-water boundaries

(Figure 1, yellow lines (Gerrish et al., 2021)). The tidewater glaciers that form these frontiers are the outer regions of two large icefields, the Warsaw and Krakow Icefields. Both icefields are reportedly experiencing unprecedented transformation due to the effects of climate warming (Rückamp et al., 2010; Dziembowski and Bialik, 2022) and are draining into AB through numerous glacial creeks.

    To summarize this dataset, it was decided to distinguish different zones within AB to recognize distinctions in water properties

that were dependent on proximity to the glacial fronts and open ocean waters. Three zones have been assigned:

-    Glacial coves: Distinct smaller bays formed near tidewater glaciers in which marine waters are under the direct influence of glacial meltwater input. Here, three glacial coves were analysed in depth, the cove near Lange Glacier (1.50 km$^2$ in area with a 2.81 km long ice-water frontline), Spera Cove (2.45 km$^2$ in area with a 4.33 km long ice-water frontline) near Vieville Glacier and Suszczewski Cove near Ecology Glacier (0.69 km$^2$ in area with a 0.36 km
55        long ice-water frontline). All three of these basins are undergoing long-term transformation caused by continuously moving and developing glacial fronts. This is visualized in Figure 1, where the light blue line on the glacial cove insets represents the ice-water boundary in 2018 (based on a Sentinel satellite image from Mar 10$^{th}$, 2018), which is different from the frontline shown on a satellite picture presented in Figure 1 taken in December of 2021 (Sentinel,



Dec 29, 2021). The change is especially noticeable in Spera Cove near Vieville glacier, where the ice front has
retreated 500 m within three years in some locations.

-      Main body of Admiralty Bay: Open bay waters in the main body of the cove, most directly influenced by the open
       ocean waters of the Bransfield Strait with which it is connected by a 13.45 km wide outlet. Nevertheless, this location
       is also affected by glacial input, especially in its northern parts.

-      Ezcurra Inlet (area of 21.32 km$^2$): This is an intermediate area separated from Admiralty Bay waters by a relatively
narrow passage (2.40 km wide) and influenced by the surrounding ice coastline (9.58 km long of 32.67 km long
       coastline).

These areas are shown in Figure 1 and are used as separate, but deeply interrelated, regions for further study. To that end,
measurement points were chosen, whose localizations are marked on the map in Figure 1, and their details (location, depth,
number of measurements performed at a given point, and, in the case of glacial cove points, distance from the water-ice
boundary) are summarized in Table 1.

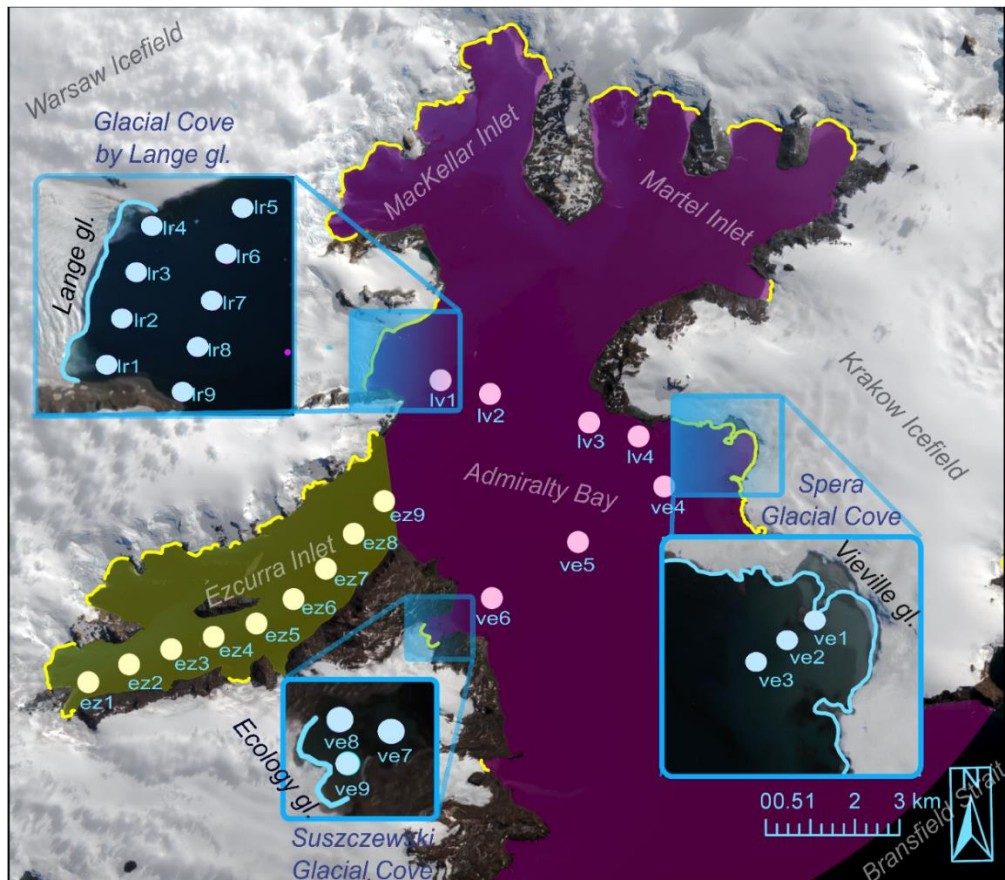

**Figure 1: Map of Admiralty Bay with measurement points in three distinct zones: main Admiralty Bay (pink), Glacial coves (blue inset boxes) and Ezcurra Inlet (yellow lines based upon Gerrish et al., 2021). Bright yellow shows current position of the ice-water coastline, in bright blue insets the position of the coastline on Mar 10$^{th}$, 2018. Source: Sentinel imagery, Dec 29$^{th}$, 2021.**



**Table 1: Details of the measurement sites**

| site name and zone | latitude | longitude | depth (m) | distance from glacial front (m) (2018-2021) |
|---|---|---|---|---|
| **Glacial Coves** | | | | |
| **lr1** | -58.4892 | -62.1227 | 19 | 315-322 |
| **lr2** | -58.4868 | -62.1195 | >100 | 266-330 |
| **lr3** | -58.4845 | -62.1163 | >100 | 275-332 |
| **lr4** | -58.4821 | -62.1131 | 23 | 260-343 |
| **lr5** | -58.4687 | -62.1120 | 8 | 940-1018 |
| **lr6** | -58.4711 | -62.1152 | 66 | 880-951 |
| **lr7** | -58.4734 | -62.1184 | >100 | 902-952 |
| **lr8** | -58.4758 | -62.1216 | >100 | 868-912 |
| **lr9** | -58.4782 | -62.1247 | 3 | 929-932 |
| **ve1** | -58.3380 | -62.1361 | 2 | 71-481 |
| **ve2** | -58.3429 | -62.1375 | 8 | 359-780 |
| **ve3** | -58.3483 | -62.1391 | 29 | 686-1118 |
| **ve7** | -58.4613 | -62.1716 | 4 | 455-469 |
| **ve8** | -58.4677 | -62.1709 | 2 | 210-232 |
| **ve9** | -58.4668 | -62.1734 | 3 | 113-128 |
| **Admiralty Bay** | | | | |
| **lv1** | -58.4624 | -62.1221 | >100 | |
| **lv2** | -58.4412 | -62.1251 | >100 | |
| **lv3** | -58.3989 | -62.1313 | 71 | |
| **lv4** | -58.3777 | -62.1343 | 17 | |
| **ve4** | -58.3671 | -62.1445 | 58 | |
| **ve5** | -58.4047 | -62.1553 | >100 | |
| **ve6** | -58.4424 | -62.1662 | 55 | |
| **Ezcurra Inlet** | | | | |
| **ez1** | -58.6172 | -62.1812 | 56 | |
| **ez2** | -58.5994 | -62.1778 | 67 | |
| **ez3** | -58.5811 | -62.1750 | 51 | |
| **ez4** | -58.5626 | -62.1727 | 61 | |
| **ez5** | -58.5441 | -62.1702 | 68 | |
| **ez6** | -58.5279 | -62.1655 | 84 | |
| **ez7** | -58.5136 | -62.1595 | >100 | |
| **ez8** | -58.5012 | -62.1526 | >100 | |

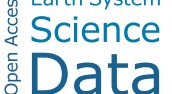

| ez9 | -58.4878 | -62.1462 | >100 |

Measurements in Glacial Coves and Admiralty Bay were taken from December 2018 until January 2022 and in Ezcurra Inlet from October 2019 till January 2022.

## 3 Methodology

### 3.1 Measured water properties


Measurements were performed with two professional YSI multiparameter EXO sondes (EXO1 and EXO2), designed for simultaneous investigation of multiple water quality properties, used and tested by researchers worldwide (Snazelle, 2015). EXO1 consists of five sensor ports and EXO2 contains seven ports; therefore, the water properties measured varied between the particular campaigns. Of the 3045 measurements collected, 2069 were acquired using the EXO1 sonde, and the remaining

976 were acquired with EXO2 and its larger sensor capacity (details seen in Figure 2).

The list of sensors and properties investigated by them is summarized in Table 2. Some hydrographic properties are derived indirectly from the direct measurements of other water column properties. In these cases, the sondes automatically calculated the additional related values based on universally accepted formulas. For example, salinity was calculated based on conductivity and temperature measurements (for details see YSI manual (YSI Inc, 2017)).

**Table 2. List of sensors and measured water properties (based upon (YSI Inc, 2017))**

| | | Sensor | Measured property | Unit | Accuracy/linearity |
|---|---|---|---|---|---|
| EXO2 | EXO1 | Conductivity/Temperature | Conductivity | µS/cm | 0-100 mS/cm: ±0.5% of |
| | | | Specific | µS/cm | reading or 0.001 mS/cm, |
| | | | Conductivity | | whichever is greater; |
| | | | nLF Conductivity | µS/cm | 100-200 mS/cm: ±1% of |
| | | | Salinity | PSU | reading |
| | | | Temperature | °C | ± 0.01° |
| | | Depth and Level | Pressure | PSI | ± 0.04 m |
| | | | Depth | m | |
| | | Dissolved Oxygen | Dissolved Oxygen | mg/L | ± 0.01 mg/L |
| | | | Dissolved Oxygen Saturation | % | |
| | | | Dissolved Oxygen Local Saturation | % | |
| | | pH | pH | -, mV | ± 0.01 |





| | | | | |
|---|---|---|---|---|
| | Turbidity | Turbidity | FNU | 0.3 FNU or ±2% of reading, whichever is greater |
| not measured by EXO1 | fDOM | Dissolved Organic Matter | QSU, RFU | $R^2>0.999$ for serial dilution of 300 ppb Quinine Sulphate solution |
| | Total Algae (Chl & BGA) | Chlorophyll A | µg/L, RFU | $R^2>0.999$ for serial dilution of Rhodamine WT solution from 0-400 µg/L Chl equivalents |
| | | BGA PE (Phycoerythrin) | µg/L, RFU | $R^2>0.999$ for serial dilution of Rhodamine WT solution from 0-280 µg/L PE equivalents |

## 3.2 Measurement procedure and sources of possible data errors and missing values

Measurements were conducted from the deck of the Zodiac boat (Figure 2). When the boat was in the designated point, the sonde lowered by the cable from the reel to a maximum depth of 100 m. At sites where the depth surpasses 100 m, the cable

was fully deployed, and at shallower points, the instruments was lowered until the depth indicator on the handheld device stabilized. This shows the limitation of this study as data was not obtained from bottom portions of the water column, particularly in the central part of the bay (see Table 1 for information on which sites bottom layers have not been investigated). The sampling rate of the sondes were initially 0.2 Hz up to December 30th, 2019, when the sampling frequency increased to 1 Hz.

The intended descent rate of the instrument was 1 m per second, but since this was manually controlled by the research personnel, the descent rate of the sonde varied significantly. Furthermore, the fact that the measurements were acquired by different crews may cause some discrepancy in the acquired data. The sensitivity of particular sensors varied, meaning that if the probe was lowered too quickly the measurements taken by some sondes may incorrectly correlate with the depth attributed to those measurements.

Other obstacles were caused by challenging weather and sea conditions. Often, waving and surface currents considerably influenced the position of the boat, making it impossible to remain stationed at the assigned site location for the duration the cast. This can be seen by position data recorded via handheld GPS during sensor deployment and included within the data file. Currents below the surface moved the sonde and cable horizontally from the initial cast position by an unknown extent.

On numerous occasions ice prevented scientists from reaching specific sites. This was frequently the case in areas close to
glacial fronts, most notably when the water surface froze during winter months and when glacial calving increased in the
summer.

All of the sensors were calibrated in accordance with guidelines found in the YSI EXO Manual (YSI Inc, 2017) and replaced
after the appropriate time or when malfunctions occurred that could not be otherwise resolved. The depth/pressure level sensor
was calibrated at the start of every survey day.

Optical sensors for turbidity, total algae and fDOM were calibrated using deionized water. However, the correct standards
occasionally showed negative values. This was particularly frequent for measurements of phytoplankton pigments. These
values have been left intact in the data file since they represent the correct variability of these properties; however, their
absolute values should be considered carefully, and more attention should be given to the relative units (RFU) for chlorophyll
A, phycoerythrin and fDOM. Turbidity FNU values have been confirmed in Admiralty Bay waters through the laboratory
procedure explained in detail by Wójcik-Długoborska et al. (2022).

Finally, after basic data analysis, some extreme outlier data, most likely caused by incidental crew mistakes or temporary sonde
malfunctions, were manually removed from the final dataset.

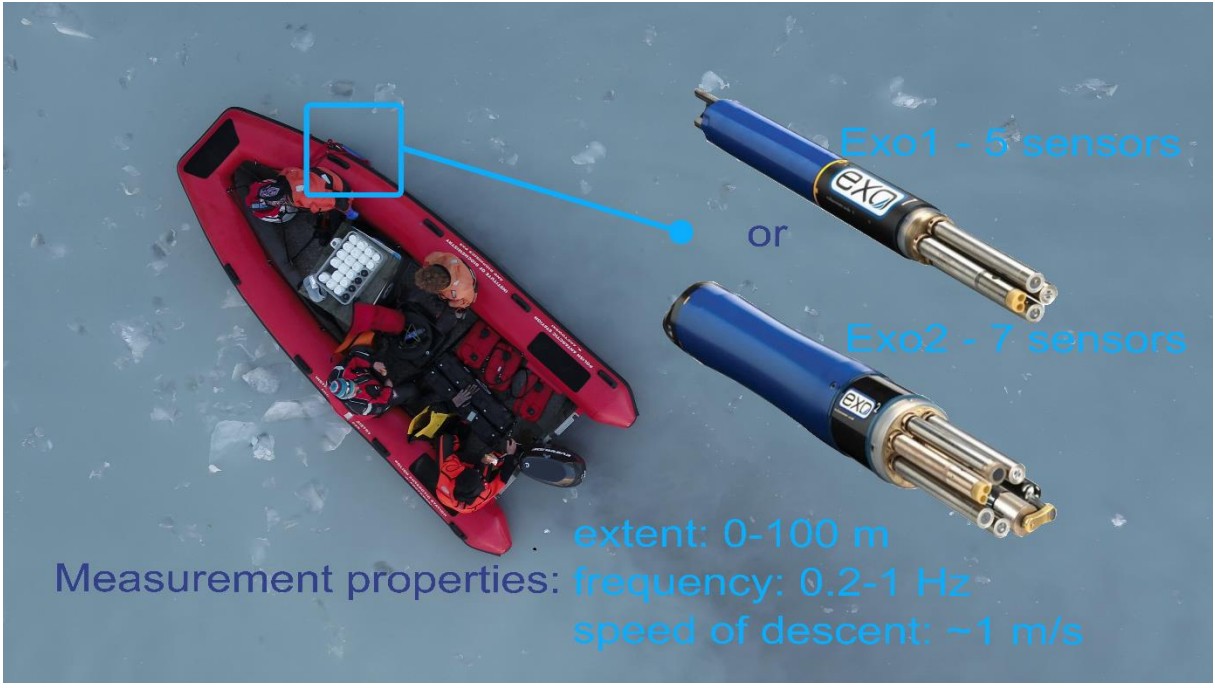

**Figure 2. Measurements visualisation, used sondes and measurement properties. In the background scientists in Zodiac boat during**
**measurements of water properties visibly infused with turbid GMW (source of sonde close-ups: *https://observator.com/products/ysi-**
***exo-series-multiparameter-sonde/*)**





# 4 Results

The results of the measurement campaign discussed above consist of a large and complex dataset describing the variability of the physical, chemical and biological properties in glacially influenced bays. Figure 3 presents a summary of the total number

of investigations performed. This shows that even at the sites sampled the least, it was possible to gather data during all seasons. However, most studies were performed during summer across all zones, while the fewest measurements were collected in winter. Interestingly, despite the unpredictable conditions in the Glacial coves, the number of surveys at each site fluctuates around one hundred per location (average 98.2 measurements per site), which is promising for future statistical analysis.

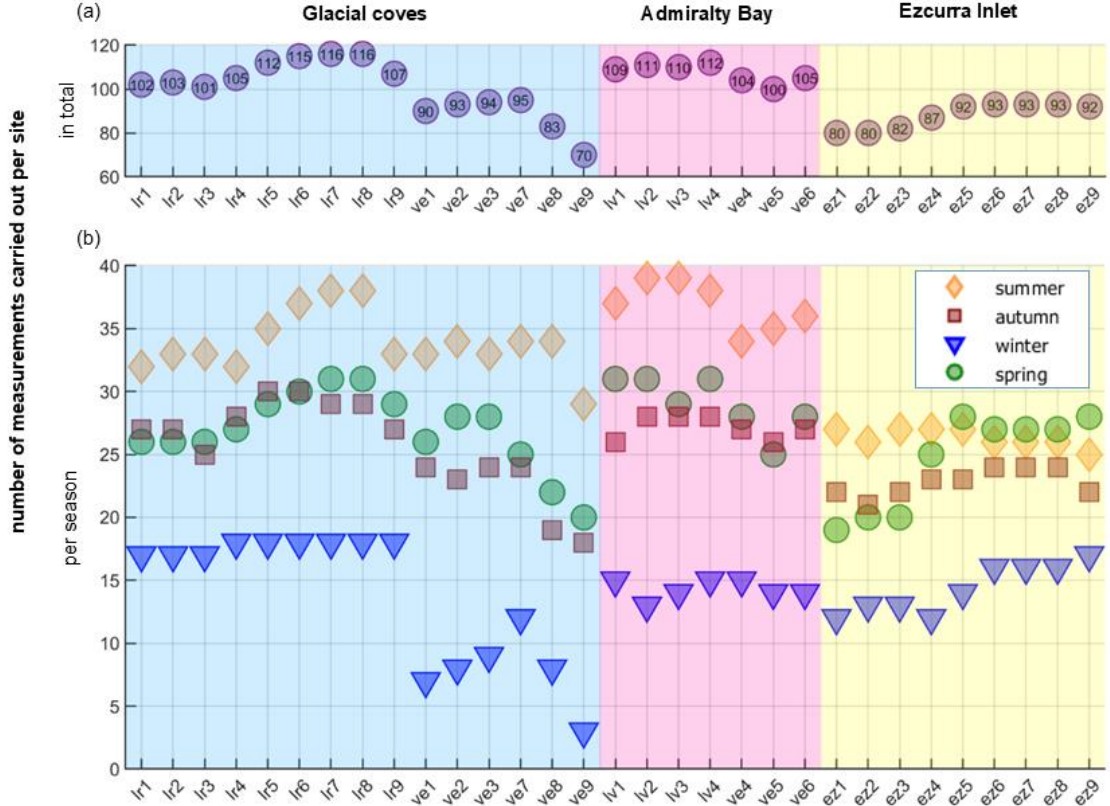

**Figure 3. Number of measurements taken at the designated sites in total (a) and by season (calendar) (b) in the period from December 2018 until January 2022**

Considering the complete duration of the projects (see Figure 4), it is noticeable how the number of measurement days fluctuated, with increases during the warmer seasons where there was a maximum of seven measurement days per month. In Figure 4(b), we observe that the same tendencies apply to all the zones, and none of them have been more frequently

investigated to any degree of significance. The average number of measurements per month was 3.74 in the Glacial coves and

2.91 in Admiralty Bay, with the same number of successful measurement days (111) throughout the whole duration of the project, and 2.42 for Ezcurra Inlet over 92 measurement days.

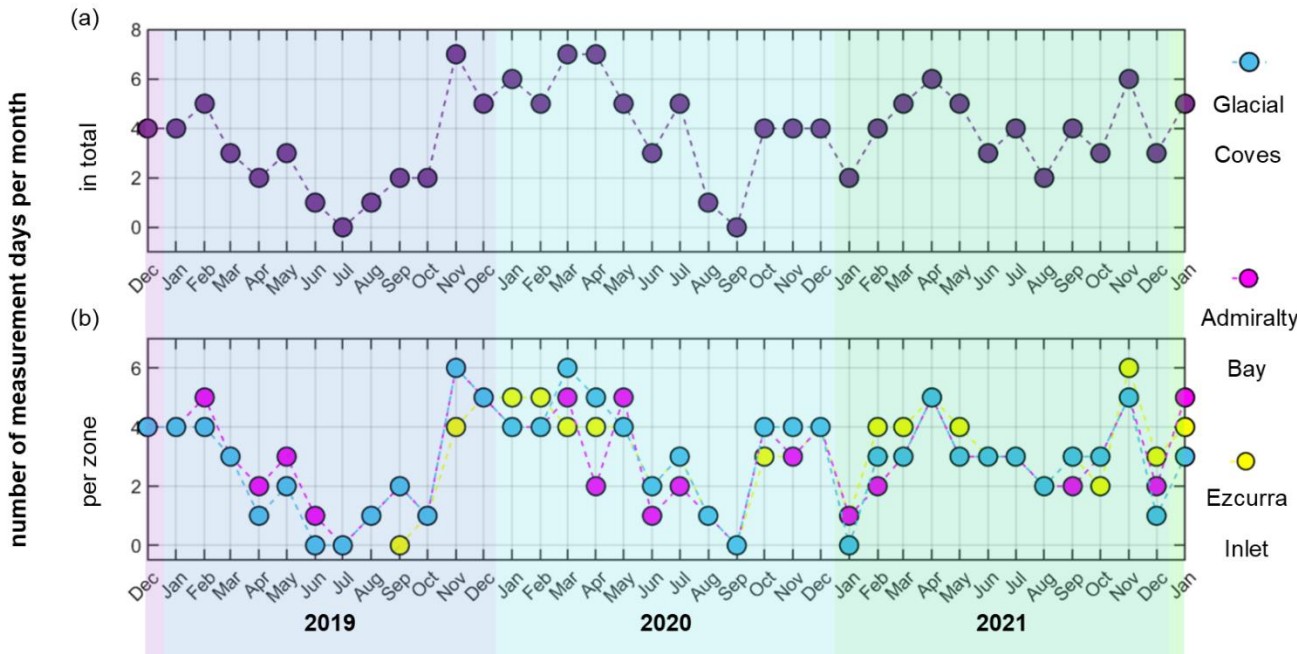

**Figure 4. Number of measurement days per month, (a) in total and (b) per each zone (counted days in which measurements in at least half of zone sites have been performed)**

The division of sites into three zones shows how proximity to glacial fronts and open ocean waters alters particular water quality properties. This effect is also notably correlated with seasonal shifts (Figure 5). Figure 5 illustrates how different properties vary in surface layers in contrast with the whole column of water (limited to 100 m of depth). This shows the impact of both atmospheric forcing and glacial outflow, which, based on buoyant plume model theory (Kimura et al., 2014; Mankoff et al., 2016; Jenkins, 2011) and observations (Chauché et al., 2014; Osińska et al., 2021), is mainly contained in the top layer of the ocean. Therefore, the results provide information on seasonal changes in water properties and glacial-ocean interactions and can be used for validation of previously formulated methods of GMW tracking.

The measured mean values of fDOM, chlorophyll A and phycoerythrin during autumn and winter are negative, which demonstrates imperfections of this methodology and is due to incorrect calibration of the sensors used. This is also true for the turbidity mean monthly values shown in Figure 6. However, in Figures 5 and 6, the overall relative distribution of these parameters is in accordance with expectations, with low values for all of them seen during colder months and with turbid waters seen specifically in the top layers of the water and closer to glacial fronts. This proves the validity of hydrographic measurements as a representation of relative variability but not for information on their absolute values.






**Figure 5: Mean values of measured properties dependent on season, divided into the whole column of water and the top 5 m of surface water.**



The 38-month-long duration of the project allowed for the tracking of seasonal variability across all measured hydrographic properties and showed consistency in all cases (Figure 6). Moreover, this duration permits cautious predictions regarding long-

term shifts in water column properties and reveals the impact of climate change or other influential conditions in this region. Using more sophisticated techniques, it is possible to more precisely determine the nature of this variability. The quantities of chlorophyll A, phycoerythrin and fDOM are not presented in Figure 6 since their measurement was significantly less frequent.

**Figure 6. Boxplot of monthly properties' mean values (excluding properties measured solely by EXO2 sonde due to their**
**significantly shorter timeseries).**

**5 Data availability**

Described dataset is freely accessible at the Pangea repository via following doi: https://doi.pangaea.de/10.1594/PANGAEA.947909 (Osińska et al., 2022), under a non-restrictive license CC BY 4.0. (at this moment doi is yet inactive since the editing process in Pangea is still taking place, so for reviewing purposes please use

following link: https://www.pangaea.de/tok/068391f63c6567f178b401bcdac7ae3d3134b625).

**6 Conclusions**

The assembled dataset shared here presents an opportunity for a better understanding of Admiralty Bay water characteristics over the 38-month survey period and can be used in further studies exploring the nature and changes in glacially influenced regions in general. The measurement technique was not perfect since some optically measured parameters showed negative

values at times. However, the sheer magnitude of this investigation is validated by the 3045 separate measurements acquired on 142 separate days over 38 months and inspires optimism regarding future work and application of this data.

The scope of measured parameters (thermodynamic, physical, chemical and biological) paints a wide and precise picture of AB hydrographic variability during all months of the year and may allow for a multidisciplinary analysis of the complex processes that take place in this location. The varied settings of study sites allow for the tracking and identification of GMW

and other water masses (Straneo et al., 2011; Chauché et al., 2014). Additionally, this sizable dataset can be used as a tool for better understanding the general hydrodynamics and thermodynamics of glacial bays and fjords and may be employed for the validation of coupled glacier-ocean modelling (Cowton et al., 2015; De Andrés et al., 2021; Bertino and Holland, 2017).

**Author Contribution**

MO – conceptualization, data curation, formal analysis, investigation, methodology, validation, visualization and writing

(original draft preparation), KAW – investigation, methodology and writing (review & editing); RJB – funding acquisition, investigation, project administration, resources, supervision and writing (review & editing)

**Competing interests**

The authors declare that they have no conflict of interest.

**Acknowledgments**

This work was supported by the National Science Centre, Poland, Grant No. UMO-2017/25/B/ST10/02092 'Quantitative assessment of sediment transport from glaciers of South Shetland Islands on the basis of selected remote sensing methods'.





Calculations were made possible by software provided by CI TASK (Center of the Tri-City Academic Computer Network) in Gdańsk. Special credits are owed to all invincible members of so called MorMon team, part of Polish Antarctic Station's crew, that throughout whole period of the project, in often trying and almost always uncomfortable conditions carried out presented
measurements.

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
