# Peer review of "Annual hydrographic variability in Antarctic coastal waters infused with glacial inflow"

_Earth System Science Data, 2022_

## Author Comment (AC2)

*Figure 1: All the in-depth profiles performed during our measurements divided by season and zone*

[Figure]

*Figure 2: Histogram of values distribution of chlorophyll A, phycoerythrin and fDOM*

[Figure]

---

## Author Response (AR1)

**Authors' response to Referees' comments**

During our interactive discussion we've tried to swiftly answer Referee comments so in this document, in general, we have assembled together these comments and responses we have already posted on ESSD platform. We feel this will make it easier to follow our revision process but also, we have made some further remarks that occurred to us while correcting our manuscript. Most notably we have marked here specifically what changes we have made to the manuscript owing to Referee comments. We appreciate all the work that went into reading and reviewing our manuscript by all the Referees, and we are confident that our corrected article has been improved significantly through this process.

**1. Referee's comments:**

**a. Referee #1 comments:**

*The goodwill of the authors and the amount of work are not in doubt. The writing is good and honest, not hiding the difficulties (fDOM and chl-a measurements). If this were another region of the world ocean I would have suggested rejecting the article. Keeping in mind that the region is remote and that the information presented-though partial-may be useful for other researchers, I suggest its publication.*

**b. Referee #2 comments:**

*The manuscript by Osinska and co-authors details a long-term dataset collected in Admiralty Bay / King George Island / Antarctica. The dataset is significant as it is one of very few long-term multiparameter time series collected in polar and sub-polar regions, particularly in a coastal region. The impact of glacial meltwater and icebergs on the marine environment has long been the subject of study, and the increased export of freshwater as icebergs and*

*meltwater from the world's ice sheet and glaciers have renewed interest in the impact of this freshwater on ocean processes and dynamics. While numerous studies have been conducted in this context, these tend to be limited to single / few cruises, and usually limited to summertime when conditions allow for easier access to locations of interest. In this case, year-round human presence in Antarctic has allowed for year-round observation of coastal ocean characteristic, a difficult feat particularly in Antarctica.*

*The complete dataset is presented in this manuscript, including details regarding missing data and data quality, which provides sufficient context to use the data in subsequent research / publications. The figures and tables complement the text well, presenting sufficient details so that data coverage and quality can be assessed. Given that these datasets represents ocean profiles, it would have been nice to have a figure showing variability of properties with depth along profiles. Figure 5 does present some aspect of this variability, however.*

*The instruments used in the data collection YSI multi-parameter probes, have a history of being used in a wide array of marine applications. I am not familiar with these probes, having instead worked with other sensors. Their use in the context seems appropriate, however. I have suggested the authors add additional detail regarding data processing (which they state follows YSI protocols with a link to the manual), as I am unfamiliar with the processing pipeline for YSI instrument and would therefore as a reader benefit from knowing more about it. The authors discuss issues*

*with a small subset of the data which they trace back to calibration issues, and I've suggested the authors include additional details if available with the idea that such details may aid other researchers studying similar harsh high-latitude systems. Regardless, while absolute values for certain parameters may in some cases not be accurate, relative distributions for these parameters seem robust and reliable, making the data a valuable contribution.*

*Inspection of the data (in Ocean Data View) was facilitated by the data being accessible in tabular format in Pangea.*
*Plotting the physical data shows few (<<< 1%) unrealistic density values (a parameter not included in the dataset but derived using temperature and salinity) due to low salinity values. Similarly there is some scatter in other biogeochemical parameters, which the authors discuss. Typically hydrographic data would first be processed using manufacturer software, and then potentially subject to further QA/QC via, for example, QARTOD (https://ioos.noaa.gov/project/qartod/). I am unfamiliar with the requirements of this publication as to data quality /*
*status (e.g. raw v. L2). Adding additional details regarding processing as detailed above, here and in the Pangeo repository, would however allow for better usage of the data in the future (e.g. indicate to researchers the data should be passed through their preferred qa/qc pipeline). The authors have overall been up front about data quality in the publication, and I agree with them that the small number of unrealistic values does not impact the overall importance and robustness of their dataset.*

*Detailed feedback:*

*L 17: unrealistic instead of impossible*

*L 20: I would shorten the discussion of GMW and make it a bit more clear. For example, it's the export of freshwater that changes the ocean's chemical composition, not GMW itself (which is the resulting water mass). I suggest: "When freshwater from glaciers is introduced to the marine environments, it mixes with ambient ocean water masses leading to the formation of new glacially modified water (GMW; Straneo 2012). Freshwater export has in this way been shown to influence properties of the coastal ocean, with impacts on the hydrodynamics and thermodynamics..."*

*L30: I would change to: "While the majority of studies examining the influence of glacial meltwater on the marine ecosystem have been performed in the Northern Hemisphere, its importance for the functioning of coastal Antarctic waters has long been hypothesized (Dierssen et al. 2002)" (https://www.pnas.org/doi/10.1073/pnas.032206999).*

*L30: Here again I would say "influence of glacial meltwater export on coastal waters" instead of GMW, since GMW is primarily seawater with a component of freshwater*

*L41: West Antarctica*

*L41 / L93: One thing that isn't obvious in your description is how challenging of an environment you sampled in. I suspect the presence of ice (sea ice, bergs and bergy bits), had an impact on operations. You mention it in L110, but it would be good highlight early in the description of the place that this is a remote and harsh environment.*

*L116: how often did negative values show up? Add a % of dataset for relevant parameters. I suspect it is small, which would further demonstrate the value / robustness of the rest of your dataset*

*L153: You repeat the fact that there are negative values a few times (see L 116). You may consider consolidating that discussion when addressing the underlying reason for the negative number (say, in L 116, where you introduce it in the context of observations), and then simply note that negative values exist in the plots which is discussed earlier. Also, you mention that negative values are due to methodological and calibration issues. Do you have specific insight / recommendations into what would have corrected this issue? It may be worth including here, as it could be helpful for other scientists to know whether, for example, a seasonal calibration is needed, whether a combination of conditions (extreme cold and turbidity) reduces the accuracy of the instrument, etc. You cite the YSI manual in a number of sections, it may be good to detail some of the content here to give context to how calibration is done and what parts of this process may have been impacted in your case.*

*L175: a detail, but the link does not work as is in pdf (fine if I copy and paste into browser).*

*L179: I would remove the mention of negative values here, as you've discussed it several times prior in the text, and focus on the big picture value of your measurements. Instead, I would use sentence 1, skip 2, modify sentence 3 to highlight details of the scope of the measurements, and finish as you do. You could otherwise add a sentence as you do in the abstract, after you've summarized the strengths and value of your data, stating that while absolute values of parameters showed some issues due to calibration, the relative distribution and seasonality is still insightful, as it is one of the few existing, long term multi parameter time series in polar regions broadly.*

*Section 3.2: Additional details on data processing should be included, as the description of the data centers primarily on the collection and sensor calibration. For example: what software was used to download / record the data? Was it recorded in a YSI proprietary format, and later converted in some software? Did the profiles go through any QA/QC or interpolation / binning, as is common for seabird data processing?*

*Figure 2: Excellent idea to have a visual of observation platform and sensors, as it is a unique environment to sample in*

*Table 1: I would add details in the caption to give context to the metadata, even if some of the details appear in the text. For example: depth was measured in this way, with depth >100 indicating that... While all stations are to some extent influenced by glacial input, distance from glacial front was measured only for those stations located within designated glacial coves...*

*L265: Snazelle should be cited as "Snazelle, T.T., 2015, Evaluation of Xylem EXO water-quality sondes and sensors:*

*U.S. Geological Survey Open-File Report 2015-1063, 28 p., http://dx.doi.org/ofr20151063." as per report (i.e. including U.S. Geological Survey Open-File Report 2015-1063).*

*Citation: https://doi.org/10.5194/essd-2022-320-RC2*

### c.    Referee #3 comments:

*General comments:*

*In addition to the thorough General comments of reviewer #2 with which I entirely agree I would recommend the author to attempt an a posteriori calibration of optical measured properties in order to resolve the issue of negative values thus increasing the usability of the presented dataset in comparison with future datasets that*
*may be collected in the same area.*

*I do suggest publication of the manuscript after revisions.*

*Detailed feedback:*

*L 21 suggested text "Fjords and bays where waters mix with glacial outflow..."*

*L 23 replace "alter" with "alters"*

*L95-L96 It is not entirely clear if bottom was reached.*

*Fig. 6 The background color for years 2020 and 2021 are too similar. I different choice would be preferable.*

*Citation: https://doi.org/10.5194/essd-2022-320-RC3*

**2.    Authors' response to comments**
**a.    Authors' response to Referee#1 comments:**

We would like to thank the Referee for this comment and support for the publication of our article. We are especially grateful for recognizing that results presented here, even though not without shortcomings, have a potential for being useful for different researchers of the Antarctic region, which is certainly one of our main goals in this project.

**b.    Authors' response to Referee#2 comments**

We are very grateful for all the comments and suggestions from Referee #2. We appreciate he acknowledged the importance of our collected dataset and its rarity due to limited amount of long-term measurement campaigns in the Antarctic coastal region. We are grateful for overall positive feedback to our methodology and presentation of results, but even more so for all the questions and suggestions of improvements.

Our detailed responses to them are as follow:

*Referee #2: Given that these datasets represents ocean profiles, it would have been nice to have a figure showing variability of properties with depth along profiles. Figure 5 does present some aspect of this variability, however.*

In the initial stages of our data analysis, we actually did look closely into all the profiles in full, but we later realized that in this scale we are missing the point of the biggest variability of all the measured properties that
were noted in the surface layers. This is why we have developed Figure 5 to truly show this phenomenon, and we skipped a step of showing the reader information on profiles in their totality, to make our paper as economical as possible.  However, after reading your comment, we have realized that this information is still worth presenting, therefore we have developed Figure 4 in the revised manuscript that shows it and we have added an appropriate description of it in the text:

*Lines 164-169: In Figure 5 vertical distribution have been presented of all the gathered data. It is apparent that temperature, pH, ODO, fDOM and phytoplankton pigment values are especially prone to change due to seasonal shifts, whereas salinity and turbidity values stay similar throughout the year. However, Figure 6 in detail illustrates how different properties vary in surface layers in contrast with the whole column of water*

 *(limited to 100 m of depth), most notably in salinity and turbidity values, but true for all measured properties except for pH.*

We will have a joined response to three following comments:

*Referee #2: I have suggested the authors add additional detail regarding data processing (which they state follows YSI protocols with a link to the manual), as I am unfamiliar with the processing pipeline for YSI instrument and would therefore as a reader benefit from knowing more about it.*

 and

*Referee #2: Inspection of the data (in Ocean Data View) was facilitated by the data being accessible in tabular format in Pangea. Plotting the physical data shows few (<<< 1%) unrealistic density values (a parameter not included in the dataset but derived using temperature and salinity) due to low salinity values. Similarly there is some scatter in other biogeochemical parameters, which the authors discuss. Typically hydrographic data would*
 *first be processed using manufacturer software, and then potentially subject to further QA/QC via, for example, QARTOD (https://ioos.noaa.gov/project/qartod/). I am unfamiliar with the requirements of this publication as to data quality / status (e.g. raw v. L2). Adding additional details regarding processing as detailed above, here and in the Pangeo repository, would however allow for better usage of the data in the future (e.g. indicate to researchers the data should be passed through their preferred qa/qc pipeline).*

 and

*Referee #2: Section 3.2: Additional details on data processing should be included, as the description of the data centers primarily on the collection and sensor calibration. For example: what software was used to download / record the data? Was it recorded in a YSI proprietary format, and later converted in some software? Did the profiles go through any QA/QC or interpolation / binning, as is common for seabird data processing?*

 Thank you very much for these questions, they certainly need some more explanation and will benefit our article. The measurement data was initially recorded to YSI proprietary format, handled by the manufacturer software embedded in all of the sensors. Through this software a real-time data filtering using basic rolling filter is performed that was done using default and recommended settings of the manufacturer. To be precise, let us quote on this YSI manual:

 "As a sonde takes measurements, it compares new readings to those taken in the previous 2-30 seconds (depending on the selected option). If the new reading is not significantly different than past measurements, then it merely factors into the rolling average with older data points to create a smooth curve. If the new reading is significantly different than past measurements, then it restarts the rolling average of data points."

The default mode provides optimum data filtering with up to 40 seconds of filtering on the sensors.

 Additionally, YSI sensors perform adaptive filtering and outlier rejection, again as per YSI manual:

"The drawback to a basic rolling filter is that response time to an impulse event is delayed, and the more entries in the average summation, the longer the delay for the result to converge on the true value. To correct this, the filter algorithm monitors the new data arriving and compares it to the current averaged result, looking for indication of an impulse event. When new data deviate from the average by more than a predetermined
 tolerance, the number of data entries within the rolling average is reduced to a minimum count and the remaining values are flushed with the new data. The result is a more accurate capture of the impulse event data, entirely eliminating the inherent delay caused by the rolling average.

Every time a newly acquired data value is added, the rolling average entries are scanned for outlier data. Although such data has already been determined to fall within the tolerances defined above, the remaining worst
 offenders are removed from the rolling average calculation. This outlier rejection allows for smoother continuous data results."

Also automatically, through YSI software all the derived values were calculated from direct measurements (meaning salinity from conductivity and temperature, depth from pressure, pH from electric potential difference, quantities of turbidity, ODO, fDOM, chlorophyll A and phycoerythrin using linear regression from optical measurement results).

The gathered dataset was downloaded using YSI KorExo program from which it was downloaded into csv format that was later analyzed using Matlab. Using Matlab data quality check was performed. We have analyzed all the distribution of all the property values and extracted questionable values based on one of the following reasons:

1. Notes from the measurement crew indicated malfunctions or some difficulties during measurements
    2. In majority of measurements in sites with depth smaller than 100 m, sonde after reaching the bottom showed unrealistic values from all of the sensors which was caused by the contact with the seafloor. This was best observed through rapid spikes in turbidity values, so in all these
profiles we have cut out all the measurements, from all the sensors from before these disturbances till the end of each of affected profiles.
    3. Extreme and outlier data was scrutinized individually:
        a. Continuous abnormal values of a particular sensor during measurement day were extracted indicating sensor malfunction or decalibration
b. Incidental extreme values recorded within otherwise reasonable datasets were extracted indicating some momentary disturbance

Despite this procedure, our data did not go through any standard QA/QC procedure so we will make that clear in our revised manuscript.

To be sure the above information is given to our readers we've added another column in Table 2 (Direct
measurement (D) or calculated from other measurement (C) in which we added information from which direct measurement given values was derived. More importantly, both in Pangaea and our revised manuscript we have expended on that issue.

Specifically in Pangaea dataset abstract we have added a following extract:

*The measurement data was initially recorded to YSI proprietary format, handled by the manufacturer software*
*embedded in all of the sensors which automatically calculated derived values of properties based on direct measurements. Details on measurement sensors' calibration, resolution, accuracy and calculation formulas can be found in YSI Exo User Manual. Data was downloaded using YSI KorExo software and analyzed through Matlab in which quality check was performed – some extreme and outlier data was extracted from the dataset either because: notes from the crew indicated possible malfunctions of the equipment or other disturbances,*
*sonde reached sea bottom, or other reasons. Despite this process dataset has not gone through any standardized QA/QC procedure.*

And in our revised manuscript we have added:

*Lines 117-129: Measured data was firstly recorded in YSI proprietary format in software embedded into all of the sensors. At this stage, data went through real-time data filtering using basic rolling filter as well as adaptive*
*filtering and outlier rejection with default settings of the manufacturer (details see (YSI Inc, 2017) ). Gathered data was later downloaded using KorExo software and exported to Matlab where some outlier and extreme values have been extracted due to one of the following reasons:*

*•     Notes from the measurement crew indicated malfunctions or some other issues*

*•     On sites with depth smaller than 100 m, sonde after reaching the bottom showed unrealistic values from*
*all of the sensors which was caused by the contact with seafloor*

*•     Other extreme and outlier data was scrutinized individually:*

*    o     Continuous abnormal values of a particular sensor during measurement day were deleted indicating sensor malfunction or decalibration*

*    o     Incidental extreme values recorded within otherwise reasonable datasets were taken out*
*indicating momentary disturbances*

*Despite this series of steps whole dataset did not go through any formalized Quality Assessment or Quality Check procedure.*

We have a joined response to two of the following comments:

*Referee #2: The authors discuss issues with a small subset of the data which they trace back to calibration issues, and I've suggested the authors include additional details if available with the idea that such details may aid other researchers studying similar harsh high-latitude systems.*

and

*Referee #2: L116: how often did negative values show up? Add a % of dataset for relevant parameters. I suspect it is small, which would further demonstrate the value / robustness of the rest of your dataset L153: You repeat the fact that there are negative values a few times (see L 116). You may consider consolidating that discussion when addressing the underlying reason for the negative number (say, in L 116, where you introduce it in the context of observations), and then simply note that negative values exist in the plots which is discussed earlier. Also, you mention that negative values are due to methodological and calibration issues. Do you have specific insight / recommendations into what would have corrected this issue? It may be worth including here, as it could be helpful for other scientists to know whether, for example, a seasonal calibration is needed, whether a combination of conditions (extreme cold and turbidity) reduces the accuracy of the instrument, etc. You cite the YSI manual in a number of sections, it may be good to detail some of the content here to give context to how calibration is done and what parts of this process may have been impacted in your case.*

These are some very good points, and we appreciate them greatly, because we truly are almost certain of the main issue that caused the miscalibration of optical sensors of fDOM and Total Algae (measuring chlorophyll A and phycoerythrin content). YSI Exo manual outlies couple of procedures of calibration for the optical sensors, dependent on each sensor. These are either 1- 2- or 3-point calibrations, and obviously the more points of calibration the better the results of it. Unfortunately, we have been able to calibrate fDOM and Total Algae sensors only using 1-point procedure, using deionized water as our 0-fluorescence standard and this proved to be an insufficient method. Therefore, we've put in this information into our revised manuscript and recommend future researchers to use a more robust method of calibration.

Unfortunately, the % of negative values of the properties is not small, in fact it is 77.82% for chlorophyll A, 70.87% for phycoerythrin and 60.45% for fDOM but looking at the histograms (posted below) of their distributions and vertical profiles in the above figure we are convinced that their relative distribution is significant. We believe that the highest pick in the histograms describe the actual state of 0 for each of the properties. The only question here is with the fDOM values distribution but after further analysis it is revealed that during our measurement, we have found that in majority of the samplings the fDOM quantities were very low, suggesting lack of dissolved organic matter in the water, but when its values rose, they rose rapidly, showing sensors great sensitivity to its presence. This would explain the two spikes in fDOM values histogram with lower one showing instances of lack of dissolved organic matter in water and the second one, its variable presence.

[Figure]

[Figure]

We truly appreciate Referee #2 in depth questioning on this matter and we have added following fragment to address these issues:

*Lines: 130-139: Optical sensors for total algae and fDOM showed unrealistic negative values (77.82% of chlorophyll A, 70.87% of phycoerythrin and 60.45% of fDOM readings). This was most probably caused by*

*chosen calibration method using 1-point procedure, based on deionized water as a proxy for 0 fluorescence standard. This methodology has been outlined by the sensor manufacturer (YSI Inc, 2017), but in this environment it has proven insufficient, and suggests necessity of using more robust method of calibration for*
*future measurements. Nevertheless, these negative values have been left intact in the data file since they represent the correct variability of the properties; however, their absolute values should be considered carefully, and more attention should be given to the relative units (RFU) for chlorophyll A, phycoerythrin and fDOM.*

*Turbidity sensor also showed negative values (19.56 % of the readings), but it has been calibrated using 2-point procedure with appropriate standard and its FNU values have been confirmed in Admiralty Bay waters through*
*the laboratory procedure explained in detail by (Wójcik-Długoborska et al., 2022).*

Also, as per the Referee #2 suggestion, we have consolidated the information about our negative results in the section describing the measurement and data handling procedure and cut out its repeated mention in results and conclusion sections.

*Referee #2: L 17: unrealistic instead of impossible*

We have changed that in the revised version of the article, thank you.

*Referee #2: L 20: I would shorten the discussion of GMW and make it a bit more clear. For example, it's the export of freshwater that changes the ocean's chemical composition, not GMW itself (which is the resulting water mass). I suggest: "When freshwater from glaciers is introduced to the marine environments, it mixes with ambient ocean water masses leading to the formation of new glacially modified water (GMW; Straneo 2012).*
*Freshwater export has in this way been shown to influence properties of the coastal ocean, with impacts on the hydrodynamics and thermodynamics..."*

Thank you for that comment and suggested solution, we've used it in our revised manuscript

*Referee #2: L30: I would change to: "While the majority of studies examining the influence of glacial meltwater on the marine ecosystem have been performed in the Northern Hemisphere, its importance for the functioning of*
*coastal Antarctic waters has long been hypothesized (Dierssen et al. 2002)" (https://www.pnas.org/doi/10.1073/pnas.032206999).*

Again, thank you very much for the suggestion and reference, we've used it as well.

*Referee #2: L41: West Antarctica*

Of course, thank you for noticing, was fixed.

*Referee #2: L41 / L93: One thing that isn't obvious in your description is how challenging of an environment you sampled in. I suspect the presence of ice (sea ice, bergs and bergy bits), had an impact on operations. You mention it in L110, but it would be good highlight early in the description of the place that this is a remote and harsh environment.*

Yes, it was a challenging environment to work in. We have added following description to our location
description:

*Lines 78-82: Due to proximity to glaciers as well as harsh Antarctic weather sampling in this region was especially strenuous. Each measurement campaign lasted few hours and has been performed from the decks of small Zodiac boats (Figure 2) that provided little comfort to the crew and getting to assigned sites often involved maneuvering through moving ice packs and bits of icebergs coming from calving glaciers. Sampling during*
*winter months required working in the dark, in extremely cold temperatures with continuous contact with freezing water.*

*Referee #2: L175: a detail, but the link does not work as is in pdf (fine if I copy and paste into browser).*

Sorry for that, it was fixed.

*Referee #2: L179: I would remove the mention of negative values here, as you've discussed it several times prior in the text, and focus on the big picture value of your measurements. Instead, I would use sentence 1, skip 2, modify sentence 3 to highlight details of the scope of the measurements, and finish as you do. You could otherwise add a sentence as you do in the abstract, after you've summarized the strengths and value of your data, stating that while absolute values of parameters showed some issues due to calibration, the relative distribution and seasonality is still insightful, as it is one of the few existing, long term multi parameter time series in polar regions broadly.*

Thank you for that suggested improvement, we have cut out second sentence and modified the third and it reads as follows:

*Lines: 192-195: The assembled dataset shared here presents an opportunity for a better understanding of Admiralty Bay water characteristics over the 38-month survey period and can be used in further studies exploring the nature and changes in glacially influenced regions in general. Sheer magnitude of this investigation with 3045 separate measurements acquired on 142 different days validates its importance and inspires optimism regarding future work and application of this data.*

*Referee #2: Figure 2: Excellent idea to have a visual of observation platform and sensors, as it is a unique environment to sample in*

Thank you for that comment.

*Referee #2: Table 1: I would add details in the caption to give context to the metadata, even if some of the details appear in the text. For example: depth was measured in this way, with depth >100 indicating that... While all stations are to some extent influenced by glacial input, distance from glacial front was measured only for those stations located within designated glacial coves...*

Yes, that's a good point. We have added this information to Table 1 caption.

*Referee #2: L265: Snazelle should be cited as "Snazelle, T.T., 2015, Evaluation of Xylem EXO water-quality sondes and sensors: U.S. Geological Survey Open-File Report 2015-1063, 28 p., http://dx.doi.org/ofr20151063." as per report (i.e. including U.S. Geological Survey Open-File Report 2015-1063).*

We have corrected this in revised manuscript.

**c.  Authors' response to Referee#3 comments**

We are very grateful to Referee #3 for her/his comments, and we appreciate overall support of publication of this article. It is also valuable for us that this comment supports previous conclusions of Referee #2, with which we are also in agreement with.

Here are our direct responses to Referee's #3 comments:

*Referee #3: In addition to the thorough General comments of reviewer #2 with which I entirely agree I would recommend the author to attempt an a posteriori calibration of optical measured properties in order to resolve the issue of negative values thus increasing the usability of the presented dataset in comparison with future datasets that may be collected in the same area.*

Thank you for that suggestion and we would very much like to do a *a posteriori* calibration but unfortunately at this time it is impossible for us. One of the sondes used for this project (Exo1) unfortunately has broken down beyond repair. The Exo2 sonde is right now still in the Arctowski Antarctic station, but the process of calibration requires purchasing and shipment of specialized standards from the manufacturer in the USA to the Antarctic which can only be done by ship couple of times a year. Since our project is finishing now, we do not have resources to fund this transport and more importantly we do not have anyone from our team at the Arctowski Antarctic Station to perform this task. So, an uncomplicated endeavor in any other place in the world becomes unachievable due to remoteness of our study area. We do hope that nevertheless these optical measurements can be used by other scientists, however we acknowledge that possible future direct comparisons with other datasets would be problematic.

*Referee #3: I do suggest publication of the manuscript after revisions.*

Thank you for these words of support.

*Referee #3: L 21 suggested text "Fjords and bays where waters mix with glacial outflow…"*

Thank you for this suggestion, but we have fixed and shortened this paragraph as per suggestion of the Referee #2, so this awkward sentence was fixed.

*Referee #3: L 23 replace "alter" with "alters"*

Of course, we've altered this.

*Referee #3: L95-L96 It is not entirely clear if bottom was reached.*

The bottom was reached only at the sites in which depth was smaller than 100 meters since this was the length of
the cable used for lowering down the sonde. The information on which sites are shallower and which are deeper than 100 m can be found in Table 1, but we can see how the phrasing in these sentences could be clearer, so we have changed fragment in lines 95-96 to:

*Lines 97-100: At sites with depth smaller than 100 m (see Table 1 for information on sites' depth) the measurements were performed throughout the whole column of water, until sea bottom was reached. At sites*
*where the depth surpasses 100 m data has been collected only from top 100 m which shows the limitation of this study as data was not obtained from bottom portions of the water column.*

*Referee #3: Fig. 6 The background color for years 2020 and 2021 are too similar. I different choice would be preferable.*

We have changed our color scheme in Figure 7 (previously Figure 6) to be more distinct.

---

## Referee Report (RR1)

This is my second review of the manuscript by Osińska et al., after author's integration of a first round of reviewer comments. The authors have addressed reviewer feedback, integrating new detail regarding data and its processing that strengthen the manuscript as a whole. The dataset presents a valuable contribution. I've included some minor grammatical comments below, but otherwise deem the manuscript ready for publication.

L45: The dataset as a whole was split into three different zones within the AB, identified based on distinct seawater properties and proximity to both glacial fronts and the mouth of the bay (i.e. proximity to open ocean source waters). These include:

Table 1 caption: ... were influenced by a number of glaciers in their vicinity

L78: Due to proximity to glaciers as well as harsh Antarctic weather, ...

L90: Some hydrographic properties are derived from direct sensor measurements (e.g., turbidity from light scatter)...

L107: Often, waves and surface currents...

L117: Great additional detail

Figure 5: Great addition for summary of profiles